# Finite Element Approximations to Caputo–Hadamard Time-Fractional Diffusion Equation with Application in Parameter Identification

**Shijing Cheng [1], Ning Du [1], Hong Wang [2] and Zhiwei Yang [3,\***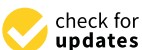**]**

[1]  School of Mathematics, Shandong University, Jinan 250100, China
[2]  Department of Mathematics, University of South Carolina, Columbia, SC 29208, USA
[3]  School of Mathematical Sciences, Fudan University, Shanghai 200433, China
**\***  Correspondence: zhiweiyang@fudan.edu.cn

**Abstract:** A finite element scheme for solving a two-timescale Hadamard time-fractional equation is discussed. We prove the error estimate without assuming the smoothness of the solution. In order to invert the fractional order, a finite-element Levenberg–Marquardt method is designed. Finally, we give corresponding numerical experiments to support the correctness of our analysis.

**Keywords:** Hadamard time-fractional diffusion equation; finite element scheme; inverted fractional order



## 1. Introduction

Fractional differential equations have been shown to provide a competitive means in modeling long-time interactions, physics and engineering, see, for instance, [1–17]. Compared to classical integer-order differential equations, the fractional equations provide a more desired descriptions of the diffusion due to the nature of the fractional operators. In particular, time-fractional partial differential equations are typically applied to model subdiffusion phenomena.

In this paper, we study a two-timescale Hadamard time-fractional diffusion equation which may describe the superslow diffusion as

$$\partial_t u + \kappa(t) {}_a^H \mathcal{D}_t^\alpha u - \Delta u = f(\boldsymbol{x}, t), \qquad (\boldsymbol{x}, t) \in \Omega \times (a, T];$$
$$u(\boldsymbol{x}, a^+) = u_a(\boldsymbol{x}), \; \boldsymbol{x} \in \Omega; \quad u(\boldsymbol{x}, t) = 0, \; (\boldsymbol{x}, t) \in \partial\Omega \times [a, T]. \tag{1}$$

Here $\partial_t u$ denotes the first derivative with respect to time, $\boldsymbol{x} := (x_1, x_2, \cdots, x_d)^\top$ and the Hadamard fractional derivative is defined by [18–20]

$$ {}_a^H \mathcal{D}_t^\alpha f(t) = \frac{1}{\Gamma(1-\alpha)} \int_a^t \left( \log \frac{t}{s} \right)^{-\alpha} f'(s) ds $$

This model can simulate strongly anomalous diffusion [21–23], which means the mean square displacement of the model has the following form

$$ <\boldsymbol{u}(t)^2> \propto \log^\mu t, \qquad \mu > 0. \tag{2} $$

An outline of our paper is as follows: We present some preliminaries and some notations in Section 2. In Section 3, we derive a finite element method for solving model problem (1) and present some numerical results to substantiate the mathematical and numerical analyses. In Section 4, we develop a finite-element Levenberg–Marquardt method to obtain the fractional order and give some numerical examples to show the utility of our method. We end this paper by giving a conclusion in the final section.

## 2. Preliminaries

We suppose that the domain $\Omega$ is a simply connected bounded domain and $\Omega \subset \mathbb{R}^d (d = 1, 2, 3)$. We introduce the following notations and the corresponding norms:

$$\|w\|_{C(\mathrm{I})} := \sup_{t \in \mathrm{I}} |w(t)|, \quad \|w\|_{C^m(\mathrm{I})} := \max_{0 \leq n \leq m} \|D^n w\|_{C(\mathrm{I})}.$$

The Sobolev space $\mathcal{H}^r(\Omega)$ with fractional index $r$ is defined in [24,25]. Let $\dot{\mathcal{H}}^2(\Omega) := \mathcal{H}^2(\Omega) \cap \mathcal{H}_0^1(\Omega)$ and $\dot{\mathcal{H}}^r(\Omega) := \left[ L^2(\Omega), \dot{\mathcal{H}}^2(\Omega) \right]_{r/2}, 0 \leq r \leq 2$, be the complex interpolation space [25]. Then, $\dot{\mathcal{H}}^0(\Omega) = L^2(\Omega)$ and $\dot{\mathcal{H}}^1(\Omega) = \mathcal{H}_0^1(\Omega)$. The definition of the fractional Laplacian is given by [26,27]

$$(-\Delta)^s w(x) = \frac{C(d, s)}{2} \int_\Omega \frac{2w(x) - w(x+y) - w(x-y)}{|y|^{n+2s}} dy, \quad s \in (0, 1), \tag{3}$$

where $C(d, s)$ is a dimensional constant. Let the operator $\mathcal{L} := -\Delta$, then the solutions $\{\varphi_i\}_{i=1}^\infty$ of the following problem

$$\begin{aligned} \mathcal{L}\varphi_i(x) &= \lambda_i \varphi_i(x), & x \in \Omega; \\ \varphi_i(x) &= 0, & x \in \partial\Omega; \end{aligned} \tag{4}$$

constitute a basis [26]. We give the definition of the fractional Sobolev spaces [10,28]

$$\check{\mathcal{H}}^r(\Omega) := \left\{ w \in L^2(\Omega) : |v|_{\check{\mathcal{H}}^r}^2 := ((-\Delta)^r w, w) = \sum_{i=1}^\infty \lambda_i^r (w, \varphi_i)^2 < \infty \right\}$$

and the corresponding norm

$$\|w\|_{\check{\mathcal{H}}^r(\Omega)} := \left( \|w\|_{L^2(\Omega)}^2 + |w|_{\check{\mathcal{H}}^\gamma(\Omega)}^2 \right)^{1/2}.$$

The space $\check{\mathcal{H}}^r(\Omega) \subset \mathcal{H}^r(\Omega)$ [10,24,28] and

$$\check{\mathcal{H}}^r(\Omega) = \left\{ w \in \mathcal{H}^r(\Omega) : (-\Delta)^k w(x) = 0, \ x \in \partial\Omega, \ k = 0, 1, \ldots, < r/2 \right\}$$

Hence $|w|_{\check{\mathcal{H}}^r(\Omega)}$ and $|w|_{\mathcal{H}^r(\Omega)}$ are equivalent in $\check{\mathcal{H}}^r(\Omega)$.

We cite some theoretical results of problem (1) in [29].

**Theorem 1.** *Suppose $\kappa(t) \in C[a, T]$, $|\kappa(t)| \leq \kappa^*$ for $t \in [0, T]$ and $\alpha \in (0, 1)$. If $u_a \in \check{\mathcal{H}}^{r+2}(\Omega)$, $f \in H^\nu(\check{\mathcal{H}}^r(\Omega))$ with $r > d/2$ and $\nu > 1/2$, then problem (1) has a solution $u$ satisfying $u \in C^1([a, T]; \check{\mathcal{H}}^r(\Omega))$ and the following holds for $G = G(\alpha, \|\kappa\|_{C[a,T]}, T)$*

$$\|u\|_{C^1([a,T];\check{\mathcal{H}}^s(\Omega))} \leq G\big( \|u_a\|_{\check{\mathcal{H}}^{2+s}(\Omega)} + \|f\|_{H^\nu(\check{\mathcal{H}}^s(\Omega))} \big), \quad 0 \leq s \leq r.$$

**Theorem 2.** *Suppose $\kappa(t) \in C[a, T]$, $|\kappa(t)| \leq \kappa^*$ for $t \in [0, T]$ and $\alpha \in (0, 1)$. If $u_a \in \check{\mathcal{H}}^{4+s}(\Omega)$, $f \in \mathcal{H}^\nu(\check{\mathcal{H}}^{2+s}(\Omega)) \cap \mathcal{H}^{1+\nu}(\check{\mathcal{H}}^s(\Omega))$ for some $s \geq 0$ and $\nu > 1/2$, $u \in C^2((a, T]; \check{\mathcal{H}}^s(\Omega))$ and the following estimate holds*

$$\|u\|_{C^2((a,T];\check{\mathcal{H}}^s(\Omega))} \leq G\left( \log \frac{t}{a} \right)^{-\alpha} \big( \|u_a\|_{\check{\mathcal{H}}^{4+s}(\Omega)} + \|f\|_{\mathcal{H}^\nu(\check{\mathcal{H}}^{2+s}(\Omega))} + \|f\|_{\mathcal{H}^{1+\nu}(\check{\mathcal{H}}^s(\Omega))} \big),$$

*with $G = G(\alpha, \|\kappa\|_{C^1[a,T]}, T)$.*

## 3. Analysis of a Finite Element Scheme

In this section, we give a finite element method for solving (1) and analyze the convergence of the method.

Given an integer $N > 0$. We discretize $[a, T]$ by $t_n := a + n\Delta t$ for $0 \le n \le N$ with $\Delta t := \frac{T-a}{N}$. First, we discretize $\partial_t u$ at $t = t_n$ by

$$\partial_t u(\boldsymbol{x}, t) = \frac{u(\boldsymbol{x}, t_n) - u(\boldsymbol{x}, t_{n-1})}{\Delta t} + \frac{1}{\Delta t} \int_{t_{n-1}}^{t_n} \partial_{tt} u(\boldsymbol{x}, t)(t - t_{n-1}) dt$$
$$=: \delta_{t_n} u(\boldsymbol{x}, t_n) + E_n(\boldsymbol{x}).$$
(5)

Next, we discretize $_a^H \mathcal{D}_t^\alpha u(\boldsymbol{x}, t)$ at time $t = t_n$ by

$$_a^H \mathcal{D}_t^\alpha u(\boldsymbol{x}, t_n)$$
$$= \frac{1}{\Gamma(1-\alpha)} \sum_{k=1}^{n} \left[ \int_{t_{k-1}}^{t_k} \frac{t_k \delta_{t_k} u(\boldsymbol{x}, t)}{(\log t_n - \log t)^\alpha} \frac{dt}{t} + \int_{t_{k-1}}^{t_k} \frac{(t \partial_t u - t_k \delta_{t_k} u)(\boldsymbol{x}, t)}{(\log t_n - \log t)^\alpha} \frac{dt}{t} \right]$$
$$=: \delta_{t_n}^\alpha u(\boldsymbol{x}, t_n) + R_n(\boldsymbol{x}).$$
(6)

where

$$\delta_{t_n}^\alpha u(\boldsymbol{x}, t_n)$$
$$:= \frac{1}{\Gamma(2-\alpha)} \sum_{k=1}^{n} t_k \left[ (\log t_n - \log t_{k-1})^{1-\alpha} - (\log t_n - \log t_k)^{1-\alpha} \right] \delta_{t_k} u(\boldsymbol{x}, t_k)$$
$$= \frac{1}{\Gamma(2-\alpha_l)} \sum_{k=1}^{n} b_{n,k} \left( u(\boldsymbol{x}, t_k) - u(\boldsymbol{x}, t_{k-1}) \right),$$

$$R_n := \sum_{k=1}^{n} R_{n,k} := \frac{1}{\Gamma(1-\alpha)} \sum_{k=1}^{n} \int_{t_{k-1}}^{t_k} \frac{(t \partial_t u - t_k \delta_{t_k} u)(\boldsymbol{x}, t)}{(\log t_n - \log t)^\alpha} \frac{dt}{t}$$

The coefficients $b_{n,k}$ for $1 \le k \le n \le N$ are given by

$$b_{n,k} := t_k \frac{(\log t_n - \log t_{k-1})^{1-\alpha} - (\log t_n - \log t_k)^{1-\alpha}}{\Delta t},$$

which have the properties [30]

$$b_{n,n} > b_{n,n-1} > \cdots > b_{n,k} > \ldots b_{n,1} > 0.$$

In order to estimate the finite element error, we introduce the following Ritz projection $\mathcal{P}_h : \mathcal{H}_0^1(\Omega) \to S_h(\Omega)$ [31] and $S_h(\Omega) \subset \mathcal{H}_0^1(\Omega)$

$$\left( \nabla(\omega - \mathcal{P}_h \omega), \nabla v \right) = 0, \quad \forall v \in S_h(\Omega), \quad \text{for } \omega \in \mathcal{H}_0^1(\Omega)$$

and the corresponding error

$$\|\omega - \mathcal{P}_h \omega\|_{L^2(\Omega)} \le G h^2 \|\omega\|_{\mathcal{H}^2(\Omega)}, \quad \forall \omega \in \mathcal{H}^2(\Omega) \cap \mathcal{H}_0^1(\Omega).$$
(7)

Let $u_n := u(\boldsymbol{x}, t_n)$. We multiply Equation (1), incorporated with (5) and (6), by $w \in \mathcal{H}_0^1(\Omega)$. We give the weak form for (1) at $t = t_n$ for $n = 1, \cdots, N$

$$\left( \delta_{t_n} u_n, w \right) + \left( \nabla u_n, \nabla w \right) + \kappa(t_n) \left( \delta_{t_n}^\alpha u_n, w \right)$$
$$= (f(\cdot, t_n), w) - k(t_n)(R_n, w) - (E_n, w).$$

We throw away the local truncation errors to arrive at a finite element scheme for (1):

$$\left( \delta_{t_n} u_n, w \right) + \left( \nabla u_n, \nabla w \right) + \kappa(t_n) \left( \delta_{t_n}^\alpha u_n, w \right)$$
$$= (f(\cdot, t_n), w) - k(t_n)(R_n, w) - (E_n, w), \quad w \in S_h, \quad n = 1, \cdots, N.$$
(8)

### 3.1. Analysis of Truncation Errors

We bound the errors $E_n$ and $R_n$ defined in (5) and (6), respectively, in this subsection.

**Theorem 3.** *Let the assumptions of Theorem 2 be satisfied. Then, we have the estimates for $n = 1, 2, \ldots, N$*

$$\|E_n\|_{L^2(\Omega)} \leq GG_0 n^{-\alpha} N^{\alpha-1}, \quad \|R_n\|_{L^2(\Omega)} \leq GG_0 n^{-\alpha} N^{\alpha-1}. \tag{9}$$

*Here, $G_0 = \|u_a\|_{\check{\mathcal{H}}^4} + \|f\|_{\mathcal{H}^\gamma(\check{\mathcal{H}}^2(\Omega))} + \|f\|_{\mathcal{H}^{1+\gamma}(L^2(\Omega))}$ for some $\gamma > 1/2$.*

**Proof of Theorem 3.** We use Theorem 2 to bound $E_n$ by

$$
\begin{aligned}
\|E_n\|_{L^2(\Omega)} &\leq \frac{GG_0}{\Delta t} \int_{t_{n-1}}^{t_n} \left( \log \frac{t}{a} \right)^{-\alpha} (t - t_{n-1}) dt \leq GG_0 \int_{t_{n-1}}^{t_n} \left( \log \frac{t}{a} \right)^{-\alpha} dt \\
&\leq GG_0 T \int_{t_{n-1}}^{t_n} \left( \log \frac{t}{a} \right)^{-\alpha} \frac{dt}{t} = \frac{GG_0 T}{1-\alpha} \left( (\log \frac{t_n}{a})^{1-\alpha} - (\log \frac{t_{n-1}}{a})^{1-\alpha} \right) \\
&= \frac{GG_0 T}{1-\alpha} \left[ \left( \log(1 + \frac{n(T-a)}{Na}) \right)^{1-\alpha} - \left( \log(1 + \frac{(n-1)(T-a)}{Na}) \right)^{1-\alpha} \right],
\end{aligned} \tag{10}
$$

We use the Taylor expansion theorem

$$\log(1 + \theta) = \theta - \frac{\theta^2}{2!} + \frac{\theta^3}{3!} + \cdots,$$

to estimate (10) by

$$
\begin{aligned}
\|E_n\|_{L^2(\Omega)} &\leq \frac{GG_0 T}{1-\alpha} \left[ \left( \log(1 + \frac{n(T-a)}{Na}) \right)^{1-\alpha} - \left( \log(1 + \frac{(n-1)(T-a)}{Na}) \right)^{1-\alpha} \right] \\
&\leq \frac{GG_0 T}{1-\alpha} \left[ \left( \frac{n}{N} \right)^{1-\alpha} - \left( \frac{n-1}{N} \right)^{1-\alpha} \right] \leq \frac{GG_0 T}{1-\alpha} n^{-\alpha} N^{\alpha-1}.
\end{aligned}
$$

Next, we bound $R_{n,1}$ in (5)

$$
\begin{aligned}
\|R_{n,1}\|_{L^2(\Omega)} &\leq \left\| \int_a^{t_1} (\log t_n - \log t)^{-\alpha} \left[ \|t \partial_t u\|_{L^2(\Omega)} + \frac{t_1}{\Delta t} \int_a^{t_1} \|\partial_t u(\cdot, s)\|_{L^2(\Omega)} ds \right] \frac{dt}{t} \right\| \\
&\leq GG_0 T \int_a^{t_1} (\log t_n - \log t)^{-\alpha} \frac{dt}{t} = \frac{GG_0 T}{1-\alpha} \left[ \left( \log(\frac{t_n}{a}) \right)^{1-\alpha} - \left( \log(\frac{t_n}{t_1}) \right)^{1-\alpha} \right] \\
&\leq \begin{cases} \dfrac{GG_0 T}{1-\alpha} \left( \log(\frac{t_1}{a}) \right)^{1-\alpha}, & n = 1, \\[2mm] \dfrac{GG_0 T}{1-\alpha} \left( \log(\frac{t_n}{t_1}) \right)^{-\alpha} \left( \log t_1 - \log a \right) \leq \dfrac{GG_0 (n-1)^{-\alpha}}{N^{1-\alpha}}, & n > 1. \end{cases}
\end{aligned}
$$

We use the fact that for $t \in [t_{n-1}, t_n]$

$$
\begin{aligned}
\|t \partial_t u - t_n \delta_{t_n} u\|_{L^2(\Omega)} &\leq \|t \partial_t u - t_n \partial_t u\|_{L^2(\Omega)} + \|t_n \partial_t u - t_n \delta_{t_n} u\|_{L^2(\Omega)} \\
&\leq |t - t_n| \|\partial_t u\|_{L^2(\Omega)} + t_n \|\partial_t u - \delta_{t_n} u\|_{L^2(\Omega)} \\
&\leq \Delta t GG_0 + GG_0 T \Delta t \|u_{tt}\|_{L^2(\Omega)} \\
&\leq GG_0 T \Delta t \left( \log(\frac{t_{n-1}}{a}) \right)^{-\alpha}
\end{aligned}
$$

to bound $R_{n,n}$ for $n \geq 2$ by

$$
\begin{aligned}
\|R_{n,n}\|_{L^2(\Omega)} &\leq G\|u\|_{C^2([t_{n-1},t_n];L^2(\Omega))}\Delta t \int_{t_{n-1}}^{t_n} (\log t_n - \log t)^{-\alpha}\frac{dt}{t} \\
&\leq GG_0\big(\log \frac{t_{n-1}}{a}\big)^{-\alpha}\Delta t\big(\log \frac{t_n}{t_{n-1}}\big)^{1-\alpha} \\
&\leq \frac{GG_0(n-1)^{-\alpha}}{N^{-\alpha}}\frac{1}{N^{2-\alpha}} \leq \frac{GG_0 n^{-\alpha}}{N^{2-\alpha-\alpha}} \leq GG_0\frac{n^{-\alpha}}{N^{1-\alpha}}.
\end{aligned}
$$

We bound $R_n$ below (5) for $n \geq 3$ as follows

$$
\begin{aligned}
\left\| \sum_{k=\lceil n/2\rceil+1}^{n-1} R_{n,k} \right\|_{L^2(\Omega)} &\leq G \sum_{k=\lceil n/2\rceil+1}^{n-1} \|u\|_{C^2([t_{k-1},t_k];L_2)}\Delta t \int_{t_{k-1}}^{t_k} (\log t_n - \log t)^{-\alpha}\frac{dt}{t} \\
&\leq GG_0\big(\log \frac{t_{\lceil n/2\rceil}}{a}\big)^{-\alpha}\Delta t \int_{t_{\lceil n/2\rceil}}^{t_{n-1}} (\log t_n - \log t)^{-\alpha}\frac{dt}{t} \\
&\leq GG_0\big(\log \frac{t_n}{a}\big)^{-\alpha}\Delta t\big(\log \frac{t_n}{a}\big)^{1-\alpha} \\
&\leq GG_0\Big(\frac{n}{N}\Big)^{-\alpha}\frac{1}{N}\Big(\frac{n}{N}\Big)^{1-\alpha} \leq \frac{GG_0}{n}\Big(\frac{n}{N}\Big)^{2-\alpha-\alpha},
\end{aligned}
$$

$$
\begin{aligned}
\left\| \sum_{k=2}^{\lceil n/2\rceil} R_{n,k} \right\|_{L^2(\Omega)} &\leq G \sum_{k=2}^{\lceil n/2\rceil} \|u\|_{C^2([t_{k-1},t_k];L^2(\Omega))}\Delta t \int_{t_{k-1}}^{t_k} (\log t_n - \log t)^{-\alpha}\frac{dt}{t} \\
&\leq GG_0 \sum_{k=2}^{\lceil n/2\rceil} \big(\log \frac{t_k}{a}\big)^{-\alpha}\Delta t(\log \frac{t_n}{t_{k-1}})^{1-\alpha} - (\log \frac{t_n}{t_k})^{1-\alpha}) \\
&\leq GG_0 \sum_{k=2}^{\lceil n/2\rceil} \big(\log \frac{t_k}{a}\big)^{-\alpha}\Delta t(\log \frac{t_n}{t_k})^{-\alpha}(\log \frac{t_k}{t_{k-1}}) \\
&\leq GG_0 \sum_{k=2}^{\lceil n/2\rceil} \frac{k^{-\alpha}n^{-\alpha}}{N^{2-\alpha-\alpha}} \leq \frac{GG_0}{n}\Big(\frac{n}{N}\Big)^{2-\alpha-\alpha}.
\end{aligned}
$$

Furthermore, we incorporate the estimates above to derive

$$
\frac{GG_0}{n}\Big(\frac{n}{N}\Big)^{2-\alpha-\alpha} = GG_0\frac{n^{-\alpha}}{N^{1-\alpha}}\frac{n^{1-\alpha}}{N^{1-\alpha}} \leq GG_0\frac{n^{-\alpha}}{N^{1-\alpha}}
$$

to obtain the estimate in (9). $\square$

**Theorem 4.** *Let $\eta(\boldsymbol{x},t) := (I - \mathcal{P}_h)u(\boldsymbol{x},t)$. Suppose that the assumptions of Theorem 2 are satisfied, then we have*

$$
\|\delta_{t_n}\eta_n\|_{L^2(\Omega)} + \|\delta_{t_n}^\alpha \eta_n\|_{L^2(\Omega)} \leq GG_0 h^2, \ \ 1 \leq n \leq N.
$$

**Proof of Theorem 4.** We apply (7) to obtain

$$
\|\delta_{t_n}\eta_n\|_{L^2(\Omega)} = \frac{1}{\Delta t}\left\|(I - \mathcal{P}_h)\int_{t_{n-1}}^{t_n} \partial_t u(\boldsymbol{x},t)dt\right\|_{L^2(\Omega)} \leq Gh^2\|u\|_{C^1([a,T];\mathcal{H}^2(\Omega))},
$$

$$
\begin{aligned}
\|\delta_{t_n}^\alpha \eta_n\|_{L^2(\Omega)} &= \frac{1}{\Gamma(2-\alpha)}\left\| \sum_{k=1}^n b_{n,k}(I - \mathcal{P}_h)\int_{t_{n-1}}^{t_n} \partial_t u(\boldsymbol{x},t)dt\right\|_{L^2(\Omega)} \\
&\leq G\Delta t h^2\|u\|_{C^1([a,T];\mathcal{H}^2(\Omega))}\sum_{k=1}^n b_{n,k} \leq Gh^2\|u\|_{C^1([a,T];\mathcal{H}^2(\Omega))}.
\end{aligned}
$$

$\square$

### 3.2. Analysis of the Finite Element Method

We give the following error estimate of the finite element method (8), which could be proved by similar techniques as in Theorem 4.3 from [17], and interested readers can see the proof for more details.

**Theorem 5.** *Let the assumptions of Theorem 2 be satisfied, then we have the following results*

$$\|u_h - u\|_{L^\infty(a,T;L^2(\Omega))} := \max_{1 \le n \le N} \|u_{h,n} - u_n\|_{L^\infty(L^2(\Omega))} \le GG_0(\Delta t + h^2).$$

### 3.3. Numerical Experiments of the Finite Element Method

We give some numerical examples to justify our numerical analysis above.

**Example 1.** *Let $\Omega = (0,1)$, $[a, T] = [1, 1.2]$, $\kappa(t) = 1$, $u_a(x) = \sin(2\pi x)$ and $f(x,t) = 0$, and select the numerical solution $\hat{u}$ with $N = 2^{10}$ and $h = 1/32$ to be a reference solution. We plot the first-order time difference quotient $\delta_{t_n} u_{h,n}(1/4, t_n)$ in Figure 1. It is clear that the temporal derivative of the solution exhibits a singular behavior near the initial time, which gets stronger as $\alpha$ increases. These findings numerically satisfy Theorem 2.*

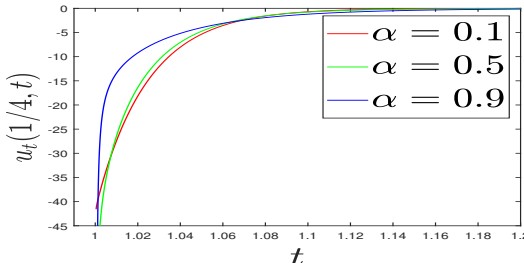

**Figure 1.** Plots of $\delta_{t_n} u_{h,n}(\frac{1}{4}, t_n)$: $\alpha = 0.1, 0.5, 0.9$.

**Example 2.** *Let $\Omega = (0,1)^2$, $[a, T] = [1, 2]$, $u_a(x) = 0$ and $\kappa(t) = 1$. The exact solution to problem (1) is chosen to be $u = (\log t)^{2-\alpha} \sin(2\pi x) \sin(2\pi y)$ and $f$ is calculated accordingly. We measure the convergence rates $\kappa$ (temporal rate) and $\gamma$ (spatial rate) by*

$$\max_{1 \le n \le N} \|u_n - u_{h,n}\|_{L^2(\Omega)} \le Q(N^{-r_t} + M^{-r_s}).$$

The numerical results listed in Tables 1 and 2 show that the finite element method retains a first-order accuracy in time and a second-order accuracy in space as proved in Theorem 5.

**Table 1.** Temporal convergence orders under different $\alpha$.

|  | $\alpha = 0.1$ |  | $\alpha = 0.5$ |  | $\alpha = 0.9$ |  |
|---|---|---|---|---|---|---|
| $N$ | $M = 2^{10}$ | $r_t$ | $M = 2^{10}$ | $r_t$ | $M = 2^{10}$ | $r_t$ |
| 32 | $4.61 \times 10^{-5}$ |  | $2.12 \times 10^{-5}$ |  | $8.64 \times 10^{-5}$ |  |
| 64 | $2.27 \times 10^{-5}$ | 1.02 | $1.11 \times 10^{-5}$ | 0.94 | $4.19 \times 10^{-5}$ | 1.04 |
| 128 | $1.13 \times 10^{-5}$ | 1.01 | $5.79 \times 10^{-6}$ | 0.94 | $2.07 \times 10^{-5}$ | 1.02 |
| 256 | $5.64 \times 10^{-6}$ | 1.00 | $2.97 \times 10^{-6}$ | 0.97 | $1.03 \times 10^{-5}$ | 1.01 |

**Table 2.** Spatial convergence orders under different $\alpha$.

|  | $\alpha = 0.1$ |  | $\alpha = 0.5$ |  | $\alpha = 0.9$ |  |
|---|---|---|---|---|---|---|
| $M$ | $N = 2^{12}$ | $r_s$ | $N = 2^{12}$ | $r_s$ | $N = 2^{12}$ | $r_s$ |
| 8 | $7.54 \times 10^{-4}$ |  | $1.55 \times 10^{-3}$ |  | $2.00 \times 10^{-3}$ |  |
| 16 | $1.93 \times 10^{-4}$ | 1.97 | $2.69 \times 10^{-4}$ | 1.97 | $5.11 \times 10^{-4}$ | 1.97 |
| 32 | $4.85 \times 10^{-5}$ | 1.99 | $6.77 \times 10^{-5}$ | 1.99 | $1.28 \times 10^{-4}$ | 2.00 |
| 64 | $1.22 \times 10^{-5}$ | 2.00 | $1.69 \times 10^{-5}$ | 2.00 | $3.10 \times 10^{-5}$ | 2.04 |

### 4. An Inversion Algorithm to Evaluate the Fractional Order $\alpha$

In many practical scenario problems, the fractional order $\alpha$ in model problem (1) is not clear. Therefore, we must use physical experiments to obtain the data.

#### 4.1. L–M Regularization Method

We discuss an algorithm to simulate the fractional order $\alpha$ with the help of the finite element method (8) as follows: given the observation data $\{\theta_i\}_{i=1}^N$, the goal of the parameter identification of $\alpha$ is to find $\alpha_{inv}$ satisfying

$$\alpha_{inv} = \arg\min_{\alpha \in (0,1)} \mathcal{F}(\alpha) := \frac{1}{2} \sum_{i=1}^N \left(u(x_i, T; \alpha) - \theta_i\right)^2. \tag{11}$$

In order to solve minimization problem (11), one can use an iterative algorithm such as Newton's method. We used the first and second derivatives of the function $\mathcal{F}(\alpha)$ for minimizing (11)

$$\alpha_{k+1} = \alpha_k - \frac{\mathcal{F}'(\alpha_k)}{\mathcal{F}''(\alpha_k)}, \tag{12}$$

Here, $k$ represents the $k$th iteration. Actually, relation (12) is equivalent to solving

$$\alpha_{k+1} = \alpha_k - (J_k^\top J_k)^{-1} J_k^\top r_k, \tag{13}$$

where $r_k = (r_1, r_2, \cdots, r_N)^\top$ and $r_i = u(x_i, T; \alpha) - \theta_i$

$$J_k = \left(\frac{\partial u(x_1, T; \alpha)}{\partial \alpha}, \cdots, \frac{\partial u(x_N, T; \alpha)}{\partial \alpha}\right)^\top \in \mathbb{R}^N. \tag{14}$$

We note that, in order to calculate the derivative $\frac{\partial u(x_i, T; \alpha)}{\partial \alpha}$, we can use the finite difference approximation

$$\frac{u(x_i, T; \alpha + \delta) - u(x_i, T; \alpha)}{\delta}$$

with a small $\delta$ to approximate the derivatives. However, the Newton method may sometimes fail to work due to $J_k^\top J_k \approx 0$. Usually, for dealing with this problem, we can apply the Levenberg–Marquardt algorithm to minimize (11) by

$$\alpha_{k+1} = \alpha_k - \left(J_k^\top J_k + \gamma_k\right)^{-1} J_k^\top r_k, \tag{15}$$

where $\gamma_k$ is a positive penalty parameter. Thus, we give the program in Algorithm 1.

---

**Algorithm 1** (A Levenberg–Marquardt Algorithm): Given the initial data, the boundary information and the observation data $\theta$.

---

1: For an initial guess $\alpha_0$ and $\rho \in (0, 1)$, $\sigma \in (0, \frac{1}{2})$, $\gamma_0 > 0$ and a $\delta \ll 1$.
2: **for** $k = 1, \cdots$, *Iterations* **do**
3:　　Solve model problem (1) corresponding to $\alpha_k$ and $\alpha_k + \delta$, respectively, to get $u_r(\cdot, T; \alpha_k)$ and $u_r(\cdot, T; \alpha_k + \delta)$.
4:　　Use formula (14) to calculate Jacobian $J_k$ and $J_k^\top r_k$ and update the search direction $d_k := -\left(J_k^\top J_k + \gamma_k\right)^{-1} J_k^\top r_k$.
5:　　Identify the search step $\rho^m$ by the following Armijo rule:

$$\mathcal{F}(\alpha_k + \rho^m d_k) \leq \mathcal{F}(\alpha_k) + \sigma \rho^m d_k J_k^\top r_k.$$

6:　　If $|\rho^m d_k| \leq$ *tolerance*, then stop and let $\alpha_{inv} := \alpha_k$. Otherwise update $\alpha_{k+1} := \alpha_k + \rho^m d_k$, $\gamma_{k+1} := \gamma_k/2$, and go to Step 3.
7: **end for**

---

### 4.2. Numerical Experiment of the Finite Element Levenberg–Marquardt Method

Next, we ran some numerical examples to test the utility of the inverse method in model problem (1). Let $\alpha^*$ be the true order of model (1). Let $\alpha_0$ be an initial guess for the optimization, $\alpha_{inv}$ be the approximation order and "Iter" be the number of iterations. We took the observation data $\theta$ to be the exact solution of the problem (1) at $T$ with fractional order $\alpha^*$.

**Test 1.** *In this test, we considered the data for the uncontaminated case.*

*Let $\kappa(t) = 1$, the exact solution can be chosen as*

$$u(x,t) = (\log t)^{2-\alpha} \sin(2\pi x),$$

*and the right-hand side can be computed accordingly. In Algorithm 1, we set $\rho = 0.75$, $\sigma = 0.25$, $\delta = 10^{-4}$ and the tolerance as $10^{-10}$. We chose $N = M = 100$ in the finite element method with different initial guesses $\alpha_0 \in (0,1)$ in the LM algorithm. We present the output $\alpha_{inv}$ and approximation error $|\alpha_{inv} - \alpha^*|$ and we also give the total number of iterations in Table 3.*

**Table 3.** Numerical observation of different $\alpha^* = 0.3, 0.6, 0.9$ with different initial guesses $\alpha_0 = 0.2, 0.5, 0.8$ in Test 1.

| $\alpha^*$ | $\alpha_0$ | $\alpha_{inv}$ | $\lvert \alpha^* - \alpha_{inv} \rvert$ | Iter. |
|:---:|:---:|:---:|:---:|:---:|
| | 0.2 | $2.9997 \times 10^{-1}$ | $3.4048 \times 10^{-5}$ | 11 |
| 0.3 | 0.5 | $2.9997 \times 10^{-1}$ | $3.3971 \times 10^{-5}$ | 11 |
| | 0.8 | $2.9997 \times 10^{-1}$ | $3.3984 \times 10^{-5}$ | 12 |
| | 0.2 | $5.9995 \times 10^{-1}$ | $5.1359 \times 10^{-5}$ | 12 |
| 0.6 | 0.5 | $5.9995 \times 10^{-1}$ | $5.1364 \times 10^{-5}$ | 11 |
| | 0.8 | $5.9995 \times 10^{-1}$ | $5.1287 \times 10^{-5}$ | 11 |
| | 0.2 | $8.9994 \times 10^{-1}$ | $5.8136 \times 10^{-5}$ | 12 |
| 0.9 | 0.5 | $8.9994 \times 10^{-1}$ | $5.8134 \times 10^{-5}$ | 12 |
| | 0.8 | $8.9994 \times 10^{-1}$ | $5.8135 \times 10^{-5}$ | 11 |

In Figures 2–4, we plotted the variation of the errors and the cost functions with the number of iterations. From the figures, we can see that the proposed finite element LM method achieved a desired approximation $\alpha^*$ for different initial guesses.

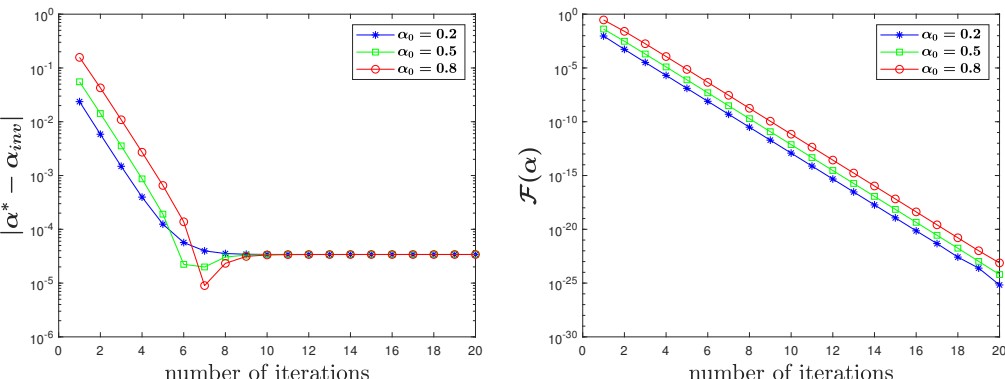

**Figure 2.** $\alpha^* = 0.3$ for uncontaminated observation data in Test 1.

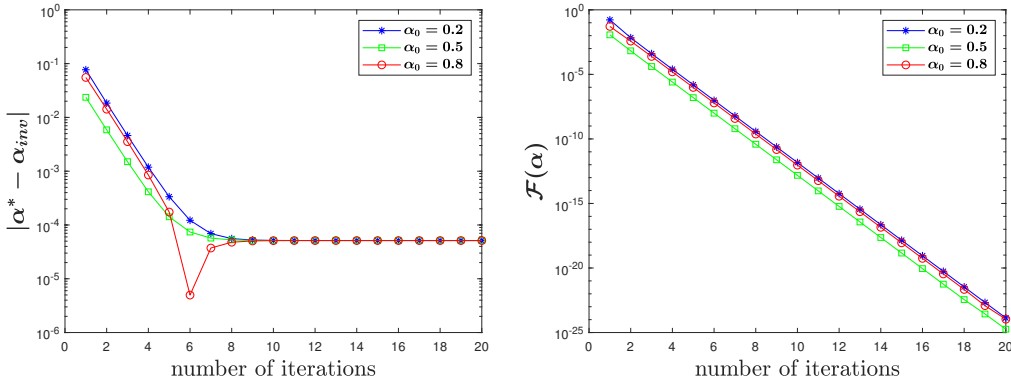

**Figure 3.** $\alpha^* = 0.6$ for uncontaminated observation data in Test 1.

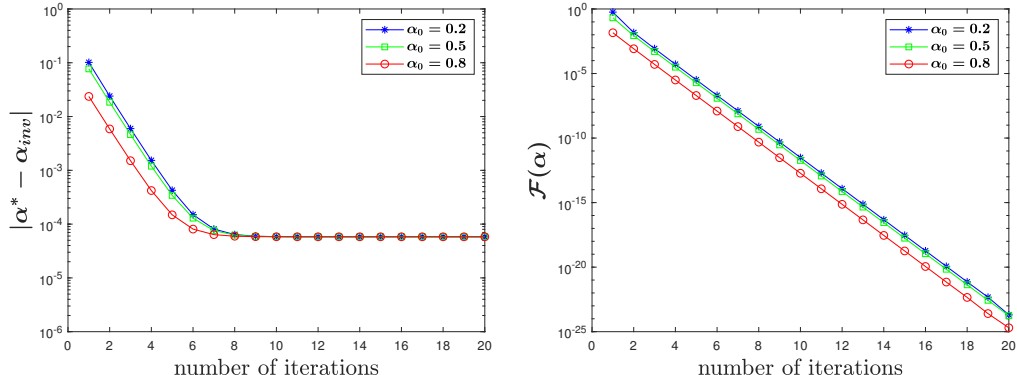

**Figure 4.** $\alpha^* = 0.9$ for uncontaminated observation data in Test 1.

**Test 2.** *Because there may be noise in real problems, we tested cases in which the data were contaminated as follows:*

$$\theta^\epsilon(x_i) = \theta(x_i)\big(1 + \epsilon\% \cdot randn(i)\big), \quad i = 1, \cdots, M. \tag{16}$$

*where $\epsilon$ represents the degree of noise, randn denotes the random noise generated by the Gaussian distribution. The exact solution, initial condition and source term in this Test were the same as those in Test 1. We listed the numerical results in Table 4.*

**Table 4.** Numerical results of different $\alpha^* = 0.75$ with $\epsilon\%$-degree noise-contaminated data in Test 2.

| $\epsilon\%$ | $\alpha_0$ | $\alpha_{inv}$ | $\lvert\alpha^* - \alpha_{inv}\rvert$ | Iter. |
|---|---|---|---|---|
| | 0.2 | $7.4999 \times 10^{-1}$ | $1.1803 \times 10^{-5}$ | 12 |
| 0.01 | 0.5 | $7.4992 \times 10^{-1}$ | $7.6152 \times 10^{-5}$ | 12 |
| | 0.8 | $7.4967 \times 10^{-1}$ | $3.3354 \times 10^{-4}$ | 12 |
| | 0.2 | $7.5078 \times 10^{-1}$ | $7.8329 \times 10^{-4}$ | 12 |
| 0.1 | 0.5 | $7.5104 \times 10^{-1}$ | $1.0446 \times 10^{-3}$ | 12 |
| | 0.8 | $7.5257 \times 10^{-1}$ | $2.5709 \times 10^{-3}$ | 12 |
| | 0.2 | $7.3639 \times 10^{-1}$ | $1.3607 \times 10^{-2}$ | 12 |
| 1 | 0.5 | $7.4047 \times 10^{-1}$ | $9.5338 \times 10^{-3}$ | 12 |
| | 0.8 | $7.4961 \times 10^{-1}$ | $3.9271 \times 10^{-4}$ | 12 |

In Figures 5–7, we also present the variation of errors and values of the cost function with iterations under different degrees of the noise. From the figures, we can see that our algorithm can obtain satisfactory results with a smaller number of iterations.

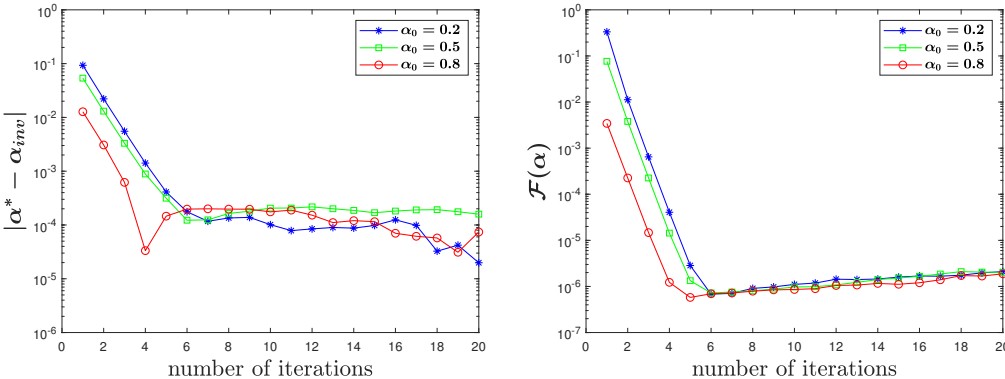

**Figure 5.** $\alpha^* = 0.75$ for contaminated observation data with 0.01% of noise in Test 2.

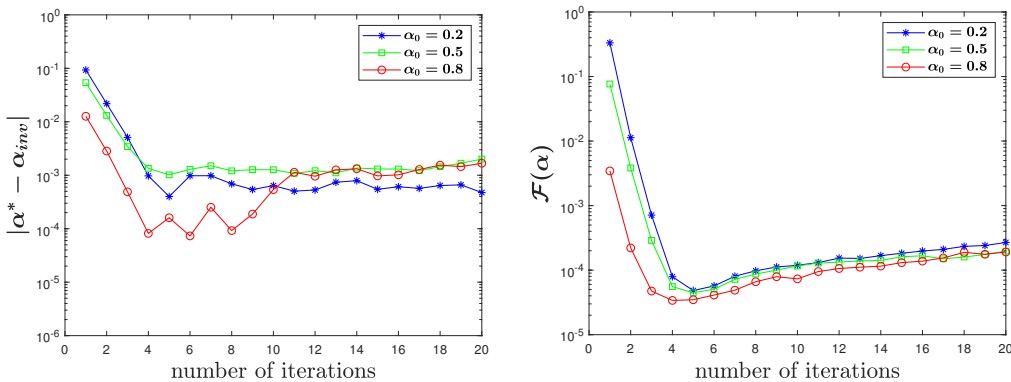

**Figure 6.** $\alpha^* = 0.75$ for contaminated observation data with 0.1% of noise in Test 2.

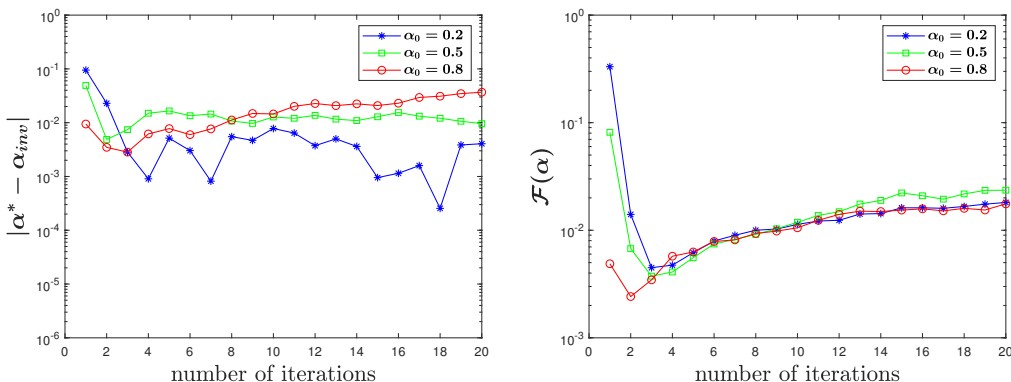

**Figure 7.** $\alpha^* = 0.75$ for contaminated observation data with 1% of noise in Test 2.

**Test 3.** *For the next test, we considered the two-dimensional case. Let $\kappa(t) = 1$, the exact solution was chosen to be*

$$u(x, y, t) = (\log t)^{2-\alpha} \sin(2\pi x) \sin(2\pi y),$$

*and the right-hand side can be computed accordingly. The other data were the same as in Test 1. We chose $N = 100$ and $h_x = h_y = 1/100$ in the finite element method with different initial guesses $\alpha_0 \in (0, 1)$ in the LM algorithm. We present the numerical results in Table 5.*

**Table 5.** Numerical results of different $\alpha^* = 0.3, 0.6, 0.9$ with different initial guesses $\alpha_0 = 0.2, 0.5, 0.8$ in Test 3.

| $\alpha^*$ | $\alpha_0$ | $\alpha_{inv}$ | $|\alpha^* - \alpha_{inv}|$ | Iter. |
|---|---|---|---|---|
| | 0.2 | $3.008 \times 10^{-1}$ | $8.4911 \times 10^{-4}$ | 11 |
| 0.3 | 0.5 | $3.008 \times 10^{-1}$ | $8.4911 \times 10^{-4}$ | 11 |
| | 0.8 | $3.008 \times 10^{-1}$ | $8.4911 \times 10^{-4}$ | 12 |
| | 0.2 | $6.008 \times 10^{-1}$ | $8.3445 \times 10^{-4}$ | 12 |
| 0.6 | 0.5 | $6.008 \times 10^{-1}$ | $8.3445 \times 10^{-4}$ | 11 |
| | 0.8 | $6.008 \times 10^{-1}$ | $8.3445 \times 10^{-4}$ | 11 |
| | 0.2 | $9.008 \times 10^{-1}$ | $8.3297 \times 10^{-4}$ | 12 |
| 0.9 | 0.5 | $9.008 \times 10^{-1}$ | $8.3297 \times 10^{-4}$ | 12 |
| | 0.8 | $9.008 \times 10^{-1}$ | $8.3297 \times 10^{-4}$ | 11 |

We also plotted the change of parameter errors and values of the cost function with iterations in Figures 8–10 and observed that the optimization process took only a few iterations to reach the tolerance.

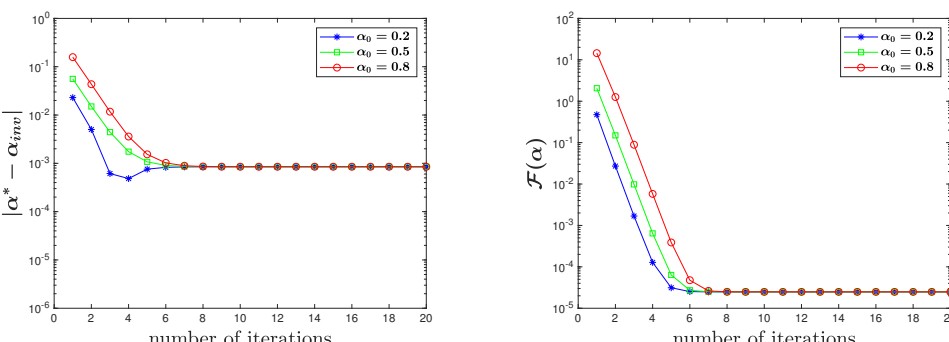

**Figure 8.** $\alpha^* = 0.3$ for uncontaminated observation data in Test 3.

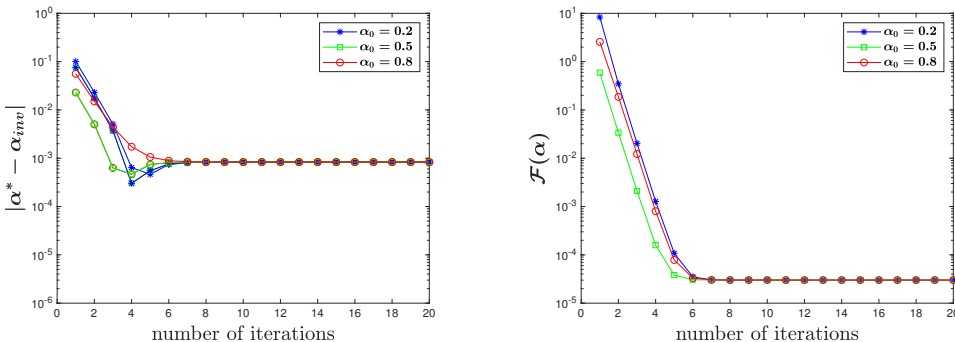

**Figure 9.** $\alpha^* = 0.6$ for uncontaminated observation data in Test 3.

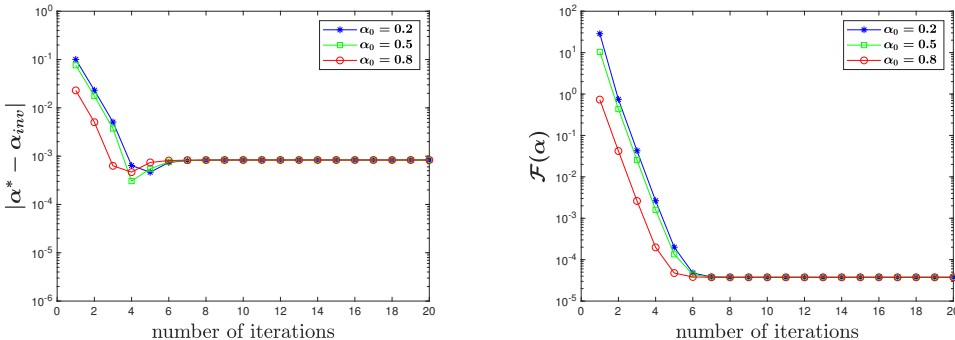

**Figure 10.** $\alpha^* = 0.9$ for uncontaminated observation data in Test 3.

**Test 4.** *We also considered two-dimensional cases in which the data had a small random perturbation as follows:*

$$\theta^\epsilon(x_i, y_j) = \theta(x_i, y_j)\big(1 + \epsilon\% \cdot randn(i,j)\big), \quad i,j = 1, \cdots, M. \tag{17}$$

*The exact solution, initial condition and source term in this Test were the same as those in Test 3. The numerical results are listed in Table 6.*

**Table 6.** Numerical observation of different $\alpha^* = 0.75$ with data contaminated with $\epsilon\%$ of noise in Test 4.

| $\epsilon\%$ | $\alpha_0$ | $\alpha_{inv}$ | $\|\alpha^* - \alpha_{inv}\|$ | Iter. |
|:---:|:---:|:---:|:---:|:---:|
| | 0.2 | $7.5085 \times 10^{-1}$ | $8.5595 \times 10^{-4}$ | 12 |
| 0.01 | 0.5 | $7.5082 \times 10^{-1}$ | $8.2265 \times 10^{-4}$ | 12 |
| | 0.8 | $7.5081 \times 10^{-1}$ | $8.1000 \times 10^{-4}$ | 12 |
| | 0.2 | $7.5089 \times 10^{-1}$ | $8.9797 \times 10^{-4}$ | 12 |
| 0.1 | 0.5 | $7.5076 \times 10^{-1}$ | $7.6190 \times 10^{-4}$ | 12 |
| | 0.8 | $7.5102 \times 10^{-1}$ | $1.0166 \times 10^{-3}$ | 12 |
| | 0.2 | $7.5415 \times 10^{-1}$ | $4.1538 \times 10^{-3}$ | 12 |
| 1 | 0.5 | $7.4991 \times 10^{-1}$ | $8.5762 \times 10^{-5}$ | 12 |
| | 0.8 | $7.5305 \times 10^{-1}$ | $3.0522 \times 10^{-3}$ | 12 |

From Figures 11–13, we can see that when the observation data $g$ are contaminated by random noise, the algorithm can still output satisfactory results in the two-dimensional case.

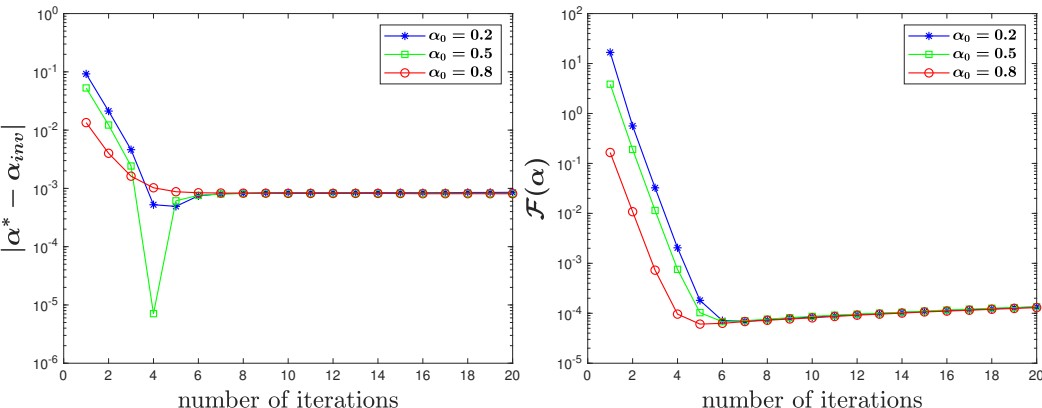

**Figure 11.** $\alpha^* = 0.75$ for true data with 0.01% of noise in Test 4.

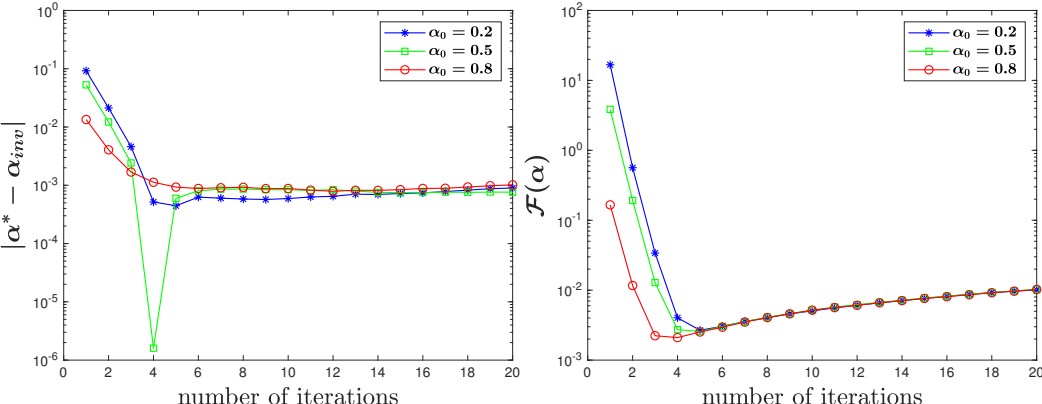

**Figure 12.** $\alpha^* = 0.75$ for true data with 0.1% of noise in Test 4.

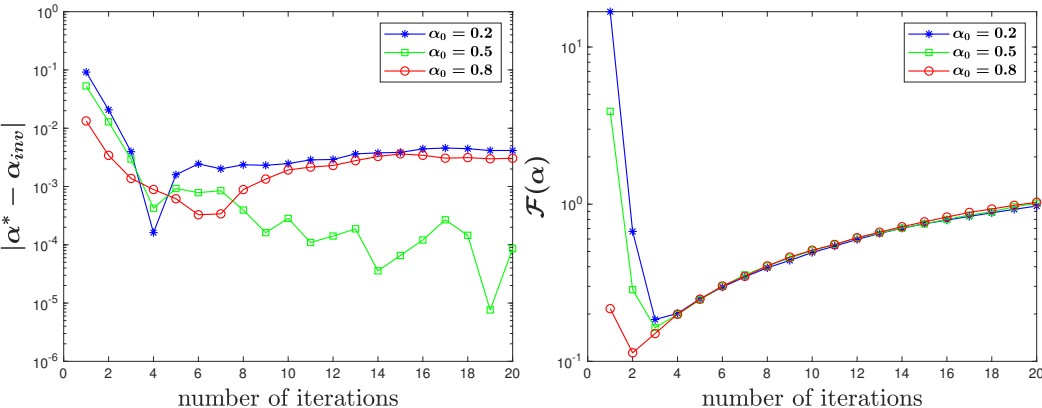

**Figure 13.** $\alpha^* = 0.75$ for true data with 1% of noise in Test 4.

## 5. Conclusions

We developed a finite element scheme for the numerical solution of two-timescale fractional diffusion Equation (1) involving a so-called Hadamard time-fractional derivative. We also proved the optimal convergence order of the error estimates of the finite element scheme to (1) without smooth assumptions on the true solution. Then, we presented several numerical examples to substantiate the mathematical and numerical analyses. We accordingly developed a finite element Levenberg–Marquardt algorithm to simulate the fractional order. In the near future, we will consider the variable-order case of the model problem (1).

**Author Contributions:** S.C.: conceptualization, methodology, validation, writing—original draft preparation; N.D.: conceptualization, writing—review and editing, funding acquisition; H.W.: conceptualization, writing—review and editing, funding acquisition; Z.Y.: conceptualization, methodology, validation, writing—original draft preparation. All authors have read and agreed to the published version of the manuscript.

**Funding:** This research was partially funded by the ARO MURI grant W911NF-15-1-0562, by the National Science Foundation under grant DMS-2012291, by the National Natural Science Foundation of China under grants 11971272, 12071262, 12131014, 11831010, and the Natural Science Foundation of Shandong Province, China (ZR2020MA048).

**Data Availability Statement:** Data sharing not applicable to this article as no dataset was generated or analyzed during the current study.

**Acknowledgments:** The authors would like to express their most sincere thanks to the referees for their very helpful comments and suggestions, which greatly improved the quality of this paper.

**Conflicts of Interest:** The authors declare no conflict of interest.

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
