# Peer review of "Finite Element Approximations to Caputo–Hadamard Time-Fractional Diffusion Equation with Application in Parameter Identification"

_fractalfract, doi:10.3390/fractalfract6090525_

Round 1

Reviewer 1 Report

Please see the attached for my comments.

Author Response

Thank you very much for your suggestions, which  improve the quality of our paper!

Reviewer 2 Report

The article entitled "Finite element approximations to Caputo-Hadamard time-fractional diffusion equation with application in parameter identification" is well written. Different proofs can be of interest for the reader in the context of non integer derivative context. I recommend for publication this work. 

Author Response

Thank you very much  for your recognition of our work!

Round 2

Reviewer 1 Report

I have checked the revised manuscript and the authors followed and improved the manuscript, thus it is satisfactory for me and I would like to accept for publication now.